# Continentalic Acid Rather Than Kaurenoic Acid Is Responsible for the Anti-Arthritic Activity of Manchurian Spikenard In Vitro and In Vivo

**DOI:** 10.3390/ijms20215488

**Published:** 2019-11-04

**Authors:** Riwon Hong, Kyoung Soo Kim, Gwang Muk Choi, Mijung Yeom, Bombi Lee, Sanghyun Lee, Ki Sung Kang, Hyang Sook Lee, Hi-Joon Park, Dae-Hyun Hahm

**Affiliations:** 1Acupuncture and Meridian Science Research Center, Kyung Hee University, Seoul 02447, Korea; smsmjhjh@naver.com (R.H.); myeom@khu.ac.kr (M.Y.); bombi@khu.ac.kr (B.L.); erc633@khu.ac.kr (H.S.L.); acufind@khu.ac.kr (H.-J.P.); 2Department of Clinical Pharmacology and Therapeutics, College of Medicine, Kyung Hee University, Seoul 02447, Korea; labrea46@naver.com; 3Department of Biomedical Sciences, Graduate School, Kyung Hee University, Seoul 02447, Korea; alsdud335@naver.com; 4Department of Plant Science and Technology, Chung−Ang University, Anseong 17546, Korea; slee@cau.ac.kr; 5College of Korean Medicine, Gachon University, Seongnam 13120, Korea; kkang@gachon.ac.kr; 6Department of Physiology, College of Medicine, Kyung Hee University, Seoul 02447, Korea; 7BioNanocomposite Research Center, Kyung Hee University, Seoul 02447, Korea

**Keywords:** continentalic acid, Manchurian spikenard, osteoarthritis, chondrocyte, monoiodoacetate

## Abstract

The aim of this study was to identify the active compound responsible for the pharmacological activities of Manchurian spikenard (*Aralia continentalis* Kitag.). Interleukin (IL)-1β-stimulated human chondrocytes and monoiodoacetate (MIA)-induced osteoarthritic rats were treated with the 50% ethanolic extract of spikenard or its major components, such as continentalic acid (ent-pimara-8(14),15-diene-19-oic acid) and kaurenoic acid (ent-kaura-16-en-19-oic acid). The spikenard extract significantly inhibited IL-1β-stimulated production of IL-6, IL-8, metalloproteinase (MMP)-1, MMP-13, cyclooxygenase (COX)-2, inducible nitric oxide synthase (iNOS) and prostaglandin(PG)E2 in a dose-dependent manner but not MMP-3 production. The extract also inhibited the IL-1β-induced translocation of NF-κB/p65 into the nucleus and dose-dependent phosphorylation levels of extracellular signal-regulated kinase (ERK), Jun amino-terminal kinase (JNK) and p38 mitogen-activated protein (MAP) kinase. Continentalic acid exhibited significant anti-arthritic activity corresponding exactly to that of the extract containing an equivalent amount of continentalic acid. On the other hand, kaurenoic acid exhibited a compatible activity at about a 10-times higher molar concentration than that of continentalic acid. In vitro anti-arthritic activities of the spikenard extract and continentalic acid were also confirmed in MIA-induced osteoarthritic rats. The 50% ethanolic extract of Manchurian spikenard exhibited promising anti-arthritic activities in the in vitro and in vivo osteoarthritis models, and continentalic acid, not kaurenoic acid, was most probably responsible for those activities.

## 1. Introduction

Non-mechanical wear and tear, as well as destructive inflammation in the joint cartilage, synovium and subchondral bones, is a relevant pathological condition causing the increase in morbidity and incident mortality in osteoarthritis (OA) patients. It is increasingly recognized that a plethora of ongoing immune responses in the cartilage and synovial membranes is a crucial manifestation in the pathogenesis and diagnosis of primary OA. Besides the inflammation in the synovial membrane (synovitis), which is a starting point of bone and cartilage destruction in the primary OA joints, the inflammation in chondrocytes in the articular cartilage is also an invaluable target for the treatment of arthritic symptoms, regardless of the arthritis type, OA or rheumatoid arthritis (RA) [1].

Until now, the most relevant medicine for OA has been analgesics, including non-steroidal anti-inflammatory drugs even though those drugs sometimes cause serious side effects such as gastrointestinal ulcer and renal morbidity. Accordingly, strategies to reduce such side effects should be considered when developing new medicines to treat OA in addition to the aim of improving the medicinal effectiveness of pain alleviation. In this respect, natural herbs and their active compounds targeting pain, inflammation or destructive joint erosion are an appropriate screening resource for isolating candidate medicines.

Manchurian spikenard (*Aralia continentalis* Kitag) is one of the 68 accepted species of the *Aralia* genus; it grows naturally in continental East Asia, including Korea and Manchuria, and is distinguished from Japanese spikenard (*Aralia cordata* Thunb.). In Korean traditional medicine, the dried roots of both spikenards have been used to treat chronic diseases accompanying pain and inflammation [2,3,4]. Several diterpenoid acids, including continentalic acid and kaurenoic acid, have been suggested to be bioactive compounds of spikenard roots with regard to anti-inflammation and analgesia [5,6,7]. However, of these constituents, the key compound responsible for their anti-inflammatory and analgesic activity has not been identified thus far.

Therefore, the current study was aimed at clearly identifying the key ingredient of the bio-active spikenard extract in terms of the anti-inflammatory, anti-nociceptive and anti-arthritic activities as well as establishing the anti-osteoarthritic effect of the Manchurian spikenard extract and its underlying mechanism in interleukin (IL)-1β-stimulated human OA chondrocytes and rat model of monoiodoacetate (MIA)-induced OA. In order to evaluate the anti-osteoarthritic effect of the spikenard extract and its chemical compounds in an in vivo animal model, cartilage degeneration was induced by intra-articular injection of MIA in the rats. MIA is an inhibitor of glyceraldehyde-3-phosphate dehydrogenase in glycolysis and thus shown to induce chondrocyte death in the articular cartilage of rodents [8].

## 2. Results and Discussion

### 2.1. Optimization of the Ethanol Content of the Extraction Solvents in RAW264.7 Cells

First, the optimal ethanol content in the extraction solvent was determined in mouse RAW264.7 macrophage cells regarding the cellular toxicity and anti-inflammatory activity of the extract. When using extraction solvents containing as much as 70% ethanol, cell viability was not affected by the treatments with the extracts up to 500 μg/mL (Appendix A). Interestingly, the 100% ethanolic extract exhibited serious inhibition of cell proliferation at 100 μg/mL, even though the contents of continentalic and kaurenoic acids were highest in 50%, not the 100%, ethanolic extraction (Appendix A). Thus, it can be assumed that the toxicity of Manchurian spikenard roots may not be attributed to continentalic or kaurenoic acid, and unidentified toxic compounds in the extract are probably more miscible in ethanol (more hydrophobic) than in those two tested compounds.

Subsequently, the anti-inflammatory and antinociceptive activities of the extracts prepared with the extraction solvents containing different content levels of ethanol were investigated in lipopolysaccharide (LPS)-treated RAW264.7 cells. Significant inhibition against LPS-stimulated expression of the biomarkers, such as IL-1β, IL-6, inducible nitric oxide synthase (iNOS) and cyclooxygenase (COX)-2, was observed at 500 μg/mL of the spikenard extract prepared with 50% or 70% ethanol, despite no inhibition at 50 μg/mL of the extract. Those inhibitory effects were more significant in the 50% ethanolic extract than the 70% or 30% extracts (Appendix A). Interestingly, these results were also coincident with the findings of the highest contents of continentalic and kaurenoic acids only in the 50% ethanolic extract, above or below which the content of those two constituents decreased (Appendix A).

### 2.2. The 50% Ethanolic Extract of Manchurian Spikenard Inhibits Interleukin (IL)-1β-Stimulated Production of Inflammatory Mediators in Human Chondrocytes

Although 50% ethanolic extract up to 200 μg/mL did not affect the chondrocyte cell viability, significant toxicity was observed with treatments above 500 μg/mL (Appendix A). Based on these results, further experiments using the chondrocytes were conducted at concentrations less than 100 μg/mL. Cell growth of the human chondrocytes was more susceptible to treatment with the 50% ethanolic extract at the same concentration than in the RAW264.7 mouse macrophages.

Following stimulation with IL-1β produced by macrophages in the synovial lining of the affected joints of arthritis patients, inflammatory mediators such as IL-6 and IL-8 are primarily secreted from chondrocytes as well as fibroblast-like synoviocytes [1,7]. In the present study, the expression levels of IL-6 and IL-8 were also markedly stimulated by IL-1β compared with those of non-treated naïve chondrocytes (none). The 50% ethanolic spikenard extract (50, 80 and 100 μg/mL) dose-dependently inhibited IL-1β-stimulated expression of those two cytokines (Figure 1A,B,D,E). The inhibition was more pronounced with IL-8 than IL-6. In the IL-8 protein production, the inhibitory effect was only significant at 100 μg/mL.

As an analgesic biomarker, the mRNA expression levels of COX-2 and its enzymatic product prostaglandin E2 (PGE2) were also investigated. The IL-1β treatment strongly activated COX-2 mRNA expression and PGE2 synthesis compared with those in non-treated naïve cells (none). Whereas IL-1β-stimulated expression of COX-2 mRNA was significantly inhibited only at 100 μg/mL of spikenard extract, the extract markedly inhibited IL-1β-stimulated PGE2 production in a dose-dependent manner (Figure 1C,F). In addition, the inhibitory effect of the extract on the mRNA expression of iNOS was investigated since NO is a more specific market of cartilage destruction in osteoarthritis. As shown in Appendix A, the extract was significantly inhibited the iNOS mRNA expression and NO production.

### 2.3. The 50% Ethanolic Extract of Manchurian Spikenard Inhibits IL-1β-Stimulated Expression of Matrix Metalloproteinases (MMPs) in Human Chondrocytes

Next, we investigated whether the 50% ethanolic spikenard extract could affect the expression levels of matrix metalloproteinases (MMPs) which perform essential roles in the irreversible destruction of joint cartilage in inflammatory arthritis. In the present study, collagenases such as MMP-1 and MMP-13 were selected as primary biomarkers of collagen degradation in the extracellular matrix (ECM) of cartilage, and MMP-3, a gelatinase, was selected as a biomarker of non-collagen matrix degradation. The inhibition of MMP activities or their expression levels in arthritic joints can be an important therapeutic strategy for the treatment of OA as well as RA [9,10,11]. As shown in Figure 2, both the mRNA and protein expression levels of MMP-1, MMP-3 and MMP-13 were markedly stimulated by pretreatment with IL-1β compared with those of non-treated naïve control (none). The stimulated expression of MMP-1 and MMP-13 was significantly inhibited by the 50% ethanolic extract of spikenard in a dose-dependent manner. Interestingly, treatment with the 50% ethanolic extract did not affect IL-1β-stimulated expression of MMP-3 at both the mRNA and protein levels (Figure 2B,E).

In our previous study, piperine, an active phenolic compound of black pepper, exhibited an inhibitory effect on MMP-13 expression, but not MMP-1, in human IL-1β-stimulated fibroblast-like synoviocytes [12]. Unlike MMP-1, which is secreted primarily by synovial membranes, MMP-13 is a product of chondrocytes and can degrade collagen and proteoglycan molecules in cartilage [13]. Because MMP-3 is secreted from chondrocytes and the synovial membrane of arthritis patients, its serum level can be a useful marker to predict cartilage damage and the treatment efficacy of antirheumatic drugs in RA patients [14,15,16]. Unlike MMP-1 and MMP-13 that can cleave the collagen triple helix directly, MMP-3 cannot cleave native collagen despite its properties of broad substrate specificity, proteoglycan degradation and activation of other MMPs. In mouse calvariae-conditioned medium, MMP-3 expression was regulated differently from MMP-13 at both the mRNA and protein levels [17]. Lee et al. also reported that estrogen reduced only the MMP-1 level, but not MMP-3, MMP-13 and TIMP-1 expression in the human chondrocyte cell line stimulated with 10 pg/mL TNF-α, a physiological level reported in the synovial fluid of OA patients [18]. These previous findings suggest that the underlying mechanism regulating MMP-3 expression can be different from those of other MMPs such as MMP-1 and MMP-13.

On the other hand, it was reported that the 70% (or 50%) ethanolic extract of Japan spikenard (*A. cordata*) showed a chondroprotective effect through the inhibition of MMP-1, MMP-3 and MMP-13 activities in IL-1α-stimulated rabbit cartilage explant cultures and a collagenase-induced inflammatory arthritis rabbit model [19,20], and that IL-1β induced a significant synthesis of IL-6, whereas IL-1α had no effect in primary neurons [21]. These results are suggesting the existence of independent signaling pathways by IL-1α and β.

### 2.4. Continentalic Acid, Not Kaurenoic Acid, is Responsible for the In Vitro Anti-Arthritic Activity of Manchurian Spikenard

Despite the wide usage of spikenards as a herbal medicine in many Asian countries, the question of which chemical component in the plant plays a key role in the pharmacological action remains unresolved. Several chemical ingredients of spikenards, such as ent-pimara-8(14),15-diene-19-oic acid (continentalic acid), 7β-hydroxy-ent-pimara-8(14),15-diene-19-oic acid, 7-oxo-ent-pimara-8(14),15-diene-19-oic acid, 15α,16α-epoxy-17-hydroxy-ent-kaurane-19-oic acid and ent-kaura-16-en-19-oic acid (kaurenoic acid) have been isolated, and their pharmacological activities have been evaluated, mainly with regard to anti-inflammatory activity [5,22]. It has previously been reported that kaurenoic acid showed analgesic, anti-inflammatory, and anti-septic activities, and that those activities were associated with the NF-κB signaling pathway [7,23,24]. The growth inhibitory effects of continentalic acid on methicillin-resistant *Staphylococcus aureus* and human liver HepG2 cancer cells were also reported [25,26,27]. However, to the best of our knowledge, there has been no report identifying the key component that can elucidate the anti-inflammatory and antinociceptive activity of continentalic acid in the Manchurian spikenard.

In order to identify the active compound being able to elucidate the anti-arthritic activity of Manchurian spikenard, two primary ingredient chemicals, continentalic and kaurenoic acids were selected as candidate compounds because the anti-inflammatory activities of those two components have been largely reported. Additionally, their in vitro anti-inflammatory activities were compared with those of the extract containing an equivalent amount of each chemical. For this, we measured the amounts of continentalic and kaurenoic acids in the extracts prepared with different ethanol solvents using high-performance liquid chromatography (HPLC, Appendix A) and it was calculated that 100 μg/mL of the 50% ethanolic spikenard extract contains 1.90 μg/mL of continentalic acid (MW: 302.50) and 0.07 μg/mL of kaurenoic acid (MW: 302.46), the concentrations of which correspond to 6.28 μM and 2.22 μM, respectively.

The cellular toxicities of continentalic and kaurenoic acids were examined in the human OA chondrocytes. In Appendix A, neither chemical exhibited any significant inhibitory effect on the cell proliferation at concentrations up to 100 μM. Both continentalic and kaurenoic acids significantly inhibited IL-1β-stimulated expression of IL-6, IL-8 and MMP-13 proteins, COX-2 mRNA, and PGE2 in a dose-dependent manner in the human OA chondrocytes (Figure 3A–F). Interestingly, the anti-inflammatory and anti-arthritic activities of continentalic acid were much stronger than those of kaurenoic acid. In Figure 3A, the inhibitory effect of 5 μM continentalic acid on IL-1β-stimulated IL-6 expression was compatible with that of 100 μg/mL of the extract in which the concentration of continentalic acid was 6.28 μM. This result indicated that the inhibitory effect of the extract on IL-1β-stimulated IL-6 expression was entirely attributable to continentalic acid. In the case of kaurenoic acid, the inhibitory activity of 100 μg/mL of the extract was close to that of 20 μM kaurenoic acid, and this concentration of kaurenoic acid is almost 10 times higher than the molar concentration (2.22 μM) of kaurenoic acid in the 100-μg/mL spikenard extract. This means that kaurenoic acid is not a key molecule responsible for the anti-inflammatory activity of Manchurian spikenard, even though kaurenoic acid has long been considered as an important medicinal compound of natural origin in the fields of medicinal chemistry and pharmacology [28]. We also observed similar patterns of continentalic acid’s superiority of anti-arthritic activity over kaurenoic acid in the in vitro experiments of other OA mediators such as IL-8, MMP-13, COX-2 and PGE2.

### 2.5. Continentalic Acid in the Spikenard Extract Inhibits IL-1β-Induced Phosphorylation of Extracellular Signal-Regulated Kinase/ Jun Amino-Terminal Kinase/p38 Mitogen-Activated Protein (ERK/JNK/p38 MAP) Kinases and Nuclear Translocation of the NF-κB/p65 Subunit

To understand the intracellular mechanism underlying anti-arthritic activity of continentalic acid, the inhibitory effects on the mitogen-activated protein (MAP) kinase signaling pathways mediated by extracellular signal-regulated kinase (ERK), Jun amino-terminal kinase (JNK) and p38 protein kinases, and NF-κB/p65 translocation, were investigated using Western hybridization (Figure 4A,B) and immunocytochemistry (Figure 4C), respectively. Continentalic acid significantly inhibited the IL-1β-stimulated phosphorylation of p38, ERK1/2, and JNK protein kinases. Furthermore, the inhibitory effects of the extract became significant at the concentrations greater than 80 μg/mL and dose-dependent for all MAP kinases (Figure 4A,B). The inhibitory effects of 50% ethanolic extract of Manchurian spikenard on all MAP kinases also showed similar patterns to those of continentalic acid (Appendix A), confirming again that continentalic acid is the genuine compound representing the anti-arthritic activity of the extract.

It has been shown that ERK1/2, JNK, and p38 protein kinases are all activated to a greater degree in OA than in normal tissue, thus playing key roles in the cartilage destruction seen in OA [21]. Therefore, the inhibitors targeting MAP kinase signaling pathways have the potential advantage of retarding disease progression and alleviating pain in osteoarthritis [29]. Of these kinases, JNK and p38 have been highlighted as therapeutic targets compared with ERK because of the lower potential toxicity of systemic inhibition and more specific response to external inflammatory signals [29].

Subsequently, the IL-1β-induced translocation of the p65 subunit of the NF-κB transcription factor was targeted as a biomarker of the inhibitory effect of continentalic acid in the inflammatory signaling pathway (Figure 4C). Pretreatment of IL-1β noticeably stimulated the nuclear translocation of the p65 subunit (red) from the cell cytoplasm into the nucleus, compared with the non-treated naïve chondrocytes (none). However, continentalic acid (10 μM) significantly inhibited the translocation of p65 in human OA chondrocytes. Taken together, these results suggested that continentalic acid in the spikenard elicited a significant suppression of the ERK1/2, JNK and p38-mediated MAP kinase signaling pathways, and resulted in a blockade of NF-κB migration into the nucleus of IL-1β-stimulated chondrocytes.

### 2.6. The Spikenard Extract and Continentalic Acid Alleviates Monoiodoacetate (MIA)-Induced Osteoarthritis in Rats

MIA-induced osteoarthritic rat model was used to examine the effects of the spikenard extract and its chemical components on alleviating pain and inflammation or cartilage erosion of arthritic knee joint (Figure 5). In the incapacitance meter test, the arthritic rats, treated with the extracts (50, 100 and 200 mg/kg), showed a significant restoration of disrupted weight balance due to the affected hindlimb, compared with vehicle-treated arthritic rats, in spite of little differences among them (Figure 5A). The effect of oral administration of 200 mg/kg spikenard extract was almost close to intraperitoneal (i.p.) treatment of 2 mg/kg indomethacin used as a positive control. The body weight restoration of the extract-treated rat groups was also supported this behavioral result, especially at the dose of 200 mg/kg (Appendix A).

We also compared anti-arthritic activity of spikenard extract (50 mg/kg, i.p.) with its chemical compounds such as continentalic and kaurenoic acids of which i.p. doses (0.97 and 0.33 mg/kg, respectively) are determined by calculating the amounts of both compounds in the extract (Figure 5B). The i,p. injections of continentalic or kaurenoic acid showed significant analgesic effects, as compared with vehicle-treated MIA group. And their analgesic effects were significantly higher than that of 50 mg/kg extract (i.p.), despite little difference between those two chemicals-treated groups. The However the pharmacological effect of continentalic acid was not significantly superior to that of kaurenoic acid at the doses equivalent to the 50 mg/kg spikenard extract, which was not coincident with the results of in vitro test using IL-1β-stimulated human chondrocytes.

Subsequently, the histochemical staining with hematoxylin-eosin (H&E) and safranin O and fast green (S-F) was used to examine proteoglycan, a major ECM component in the cartilage of knee joint (Figure 6). It was observed that the thickness (dotted black line) of articular cartilage in the MIA group was much narrower than that in the NOR group, and this cartilage defect was significantly restored in the SPIK (Manchurian spikenard extract), CONTI (continentalic acid) and KAU (kaurenoic acid) groups. Although the restoration of MIA-induced cartilage defect was not dose-dependent in the spikenard extract-treated groups, the therapeutic effects were more pronounced in continentalic acid- or kaurenoic acid-treated groups than the extract-treated groups. In the S-F staining in particular, it was clearly observed that the MIA-induced loss of proteoglycan, indicated by a color transition from red to pale pinky stain of cartilage, was significantly protected by the treatments of spikenard and its two components (Figure 6B). It was observed that the dose-dependent restoration of MIA-induced cartilage defect was significant in the Manchurian Spikenard extract-treated groups, and the continentalic acid-treated group showed the most remarkable histologic restoration of cartilage defect among all treatment groups.

Authors should discuss the results and how they can be interpreted in the perspective of previous studies and of the working hypotheses. The findings and their implications should be discussed in the broadest context possible. Future research directions may also be highlighted.

## 3. Materials and Methods

### 3.1. Cell Experiments

#### 3.1.1. Plant Material and Chemicals

The dried roots of Manchurian spikenard (*A. continentalis* Kitag.), harvested from Imsil County, South Korea, in May 2015, were supplied by the Imsil Cheese and Food Research Institute (Jeollabuk-do, Korea). The voucher specimen was deposited at the herbarium located at the Imsil Cheese and Food Research Institute (No. D201505MSI). Manchurian spikenard roots were extracted with ethanol solvent for 3 h (7 L × 3) under reflux at 65–75 °C. After filtration and removal of solvent in vacuo, the powder form of the ethanolic extract was collected (15.43% extraction yield in 50% ethanolic solvent). This extract was resuspended in distilled water for subsequent work. Kaurenoic acid (*ent*-kaura-16-en-19-oic acid) and continentalic acid (*ent*-pimara-8(14), 15-diene-19-oic acid), purchased from Sigma-Aldrich Chemical Co. (St. Louis, MO, USA) and ChemFaces (Wuhan, Hubei, China), respectively, were used as HPLC standards for the extract or chemical drugs. The HPLC chromatogram of the 50% ethanolic extract of Manchurian spikenard is shown in Appendix A.

#### 3.1.2. Cell Culture

Human OA chondrocyte primary cells, derived from the cartilage of OA patients (a 43-year-old Caucasian woman), were purchased from Cell Applications Inc. (Cat#: 402OA-05a; San Diego, CA, USA). The cells were maintained in human chondrocyte basal medium (Cat# 410-500) containing growth supplement and were purchased from Cell Applications Inc. (San Diego, CA, USA). After growing to 95% confluence for 10 days, the cells were split at a 1:4 ratio for further study. Mouse RAW264.7 macrophage cells were obtained from the Korean Cell Line Bank (Seoul, Korea). The cells were cultured in Dulbecco’s Modified Eagle’s Medium (DMEM, high glucose; Welgene, Gyeongsan-si, Korea) supplemented with 10% fetal bovine serum (Welgene), penicillin (100 U/mL), and streptomycin (100 μg/mL; Thermo Fisher Scientific, Rockford, IL, USA) at 37 °C in a humidified incubator with 5% CO_2_. In vitro experimental schedule using IL-1β-stimulated human chondrocytes and LPS-stimulated RAW264.7 cells are indicated in Appendix A.

#### 3.1.3. Reverse Transcription Polymerase Chain Reaction (RT−PCR) and Enzyme-Linked Immunosorbent Assay (ELISA)

Human OA chondrocytes (or RAW264.7 macrophage cells) with high confluency (>95%) were cultured overnight in 60-mm dishes containing 2 mL of complete medium each. Cells were incubated with serum-free medium for 24 h, and the new medium was replaced just prior to the addition of the drugs (extract, continentalic acid or kaurenoic acid) in the presence or absence of IL-1β in DMEM medium. After 6 h, total RNA was extracted from the cells using TRIzol^®^ reagent (Ambion of Thermo Fisher Scientific, Rockford, IL, USA). Total RNA (1 μg) in a 20 μL reverse transcription reaction mixture was incubated for 60 min at 42 °C using a kit for reverse transcription polymerase chain reaction (RT-PCR, TaKaRa Bio Co., Shiga, Japan). The complementary DNA was then subjected to PCR amplification with appropriate primer sets. The primer sequences and operating conditions are listed in Table 1. After amplification, the PCR products were separated on a 1 % agarose gel and were stained with GelRed^®^ (Biotium, Fremont, CA, USA). The intensity of each sample band was measured using an image analysis system (i-Max™; CoreBio System, Seoul, Korea) and compared with each other after adjusting the band intensity to that of glyceraldehyde 3-phosphate dehydrogenase (GAPDH).

For ELISA, human OA chondrocytes were seeded at 6 × 10^3^ cells/mL in six-well culture plates containing 1 mL of complete medium each and were cultivated for a week to obtain high confluency (>95%). Thereafter, culture medium was replaced with serum-free minimal medium, and the cells were cultured for another 24 h. Different concentrations (50, 80, or 100 μg/mL) of the drugs were added to the media shortly before new serum-free medium was replaced, and IL-1β (10 ng/mL) was subsequently added to stimulate the cells 1 h after the addition of the drugs. The cell supernatant was collected by centrifugation and was analyzed for IL-6, IL-8, MMP-1, MMP-3 and MMP-13 expression using ELISA kits from R&D Systems Inc. (Minneapolis, MN, USA), and for PGE2 expression using an ELISA kit from Abcam (Cambridge, UK). All assays were performed in triplicate.

#### 3.1.4. Western Hybridization

Human OA chondrocytes (6 × 10^3^ cells) were cultured in 60 mm dishes containing 4 mL of complete medium for a week to obtain high confluency (>95%). The culture medium was then replaced with serum-free minimal medium, and the cells were cultured for another 24 h. Different concentrations of the spikenard extract or continentalic acid were added to the media shortly before the new serum-free medium was replaced, and IL-1β (10 ng/mL) was subsequently added to stimulate the cells 1 h after the addition of the drugs. After 24 h of cultivation, the cells were washed twice with phosphate-buffered saline and were treated with 100 μL of lysis buffer (20 mM Tris-Cl at pH 8.0, 150 mM NaCl, and 1 M ethylenediaminetetraacetic acid (EDTA)), 1% Triton X-100, 20 μg/mL chymostatin, 2 mM phenylmethylsulfonyl fluoride, 10 μM leupeptin, and 1 mM 4-(2-aminoethyl benzenesulfonyl fluoride). The lysed samples were separated by sodium dodecyl sulfate polyacrylamide gel electrophoresis (SDS-PAGE, 12% polyacrylamide gels) and then were transferred to PVDF membrane (GE Healthcare life sciences, Freiburg, Germany). After blocking with 5% skim milk, the membranes were first incubated with various anti-rabbit polyclonal IgG for p-ERK1/2, p-P38, p-JNK, and β-actin (Santa Cruz Biotechnology Inc., Dallas, TX, USA) at 1:500 dilution in Tris-buffered saline/Tween buffer at 4 °C overnight and were further incubated at a 1:10,000 dilution of goat anti-mouse IgG secondary antibody coupled with horseradish peroxidase for probing. The membranes were subsequently developed by enhanced chemiluminescence (ECL^®^; Bio-Rad, Hercules, CA, USA).

#### 3.1.5. Immunofluorescence Microscopy

Human OA chondrocytes on glass-bottomed dishes (Corning Co., New York, NY, USA) were fixed in 100% methanol (chilled at −20 °C). After blocking with 2% bovine serum albumin, the cells were incubated with a primary antibody specific for NF-κB/p65 (1:50; Santa Cruz Biotechnology Inc., Dallas, TX, USA), followed by incubation with the Alexa Fluor 647 goat anti-rabbit immunoglobulin G secondary antibody (1:200; Molecular Probes, Thermo Fisher Scientific). Hoechst 33258 (1:500; Sigma-Aldrich Co.) was used to stain the nucleus. Images were obtained using a confocal fluorescence microscope (FLUOview FV10i; Olympus, Tokyo, Japan).

### 3.2. Animal Experiments

#### 3.2.1. Animals and Groups

Adult male SD rats weighing 180–200 g (6-week-old) were obtained from Central Lab. Animal Inc. (Seoul, Korea). The rats were housed in a limited access rodent facility at 22 ± 2 °C with up to five rats per polycarbonate cage. All animal care and experimental procedures were conducted in accordance with the National Institute of Health Guide for the Care and Use of Laboratory Animals and were approved by the Kyung Hee University Institutional Animal Care and Use Committee (KHUASP(SE)-15-115, approved in 18 March 2016). Rats were divided at random into non-treated normal group (NOR, *n* = 7), vehicle-treated MIA-arthritic group (MIA, *n* = 7), 50 mg/kg (SPIK50, *n* = 7, *p.o.* or *i.p.*), 100 mg/kg (SPIK100, *n* = 7, *p.o.*) and 200 mg/kg (SPIK200, n = 7, *p.o.*) spikenard extract-treated arthritic groups, and 0.97 mg/kg continentalic (CONTI, *n* = 7, *i.p.*) and 0.33 mg/kg kaurenoic (KAU, n = 7, *i.p.*) acid-treated MIA-arthritic group. Indomethacin (2 mg/kg) (INDO, *n* = 7, *p.o.*) was used as a positive control. The doses of continentalic and kaurenoic acids (dissolved in 5% DMSO + 3% ethanol + 92% corn oil) were determined by calculating the equivalent amounts in 50 mg/kg spikenard extract (saline). Drug treatments started 1 day after MIA (Sigma-Aldrich Chemical Co.) injection and lasted once daily for 30 days (10 a.m. everyday).

#### 3.2.2. Monoiodoacetate(MIA)-Induced Knee Arthritis

MIA-induced experimental osteoarthritis was developed in animal Lab. according to the protocol of Guingamp et al. [30]. Briefly, after being anesthetized with isoflurane in 30% O_2_/70% N_2_O, the rats were given a single intra-articular injection of 3 mg of monosodium iodoacetate (MIA; Sigma, St. Louis, MO, USA; cat #I2512) through the infrapatellar ligament of the right knee. MIA was dissolved in physiologic saline and administered in a volume of 50 μL. The amount of MIA injected into the joint was determined from a dose-response study (1, 3, 8 mg) in which the maximal degree of joint discomfort was noted using a concentration of 3 mg/joint. On day 0, the right knee joint was injected with MIA (Appendix A). For histologic studies, the left contralateral control knee was injected with 50 μL of saline and served as the control. In vivo experimental schedules using MIA-induced arthritic rats are indicated in Appendix A.

#### 3.2.3. Evaluation of Arthritic Symptoms

The arthritic pain was evaluated by the measurements of weight distribution ratio (WDR) every third day. WDR is the ratio of the per cent of weight carried on each hind leg in which the weight-bearing forces of both hind limbs were measured with an incapacitance meter (UGO-BASIL Biological Research Apparatus Co., Comerio-Varese, Italy). The bearing force of each hind limb was quantified by two mechanotransducers, separately placed below two hind legs: one was normal and the other was the arthritic leg. The bearing force of each hind leg was estimated as a 5 s average, and the mean bearing force was calculated from four separate experiments. The WDR percentage was calculated as % WDR = 100 × (weight borne by ipsilateral limb/total weight borne by both limbs).

#### 3.2.4. Histology

Knee joints in each group were randomly dissected, fixed for 5 days in 10% formalin, demineralized in Calci-Clear Rapid^TM^ solution (National Diagnotics Inc., Atlanta, GA, USA), and subsequently embedded in paraffin. The paraffin blocks were then serially sectioned at ~200 μm intervals into 8 mm-thick sections for staining using a manual rotary microtome (Finesse 325, Thermo Shandon Inc., Pittsburgh, PA, USA). The slides were stained with hematoxylin and eosin (H&E) and for general structural evaluation, and safranin O and fast green (S-F) for detection of proteoglycan in the cartilage of knee joint.

### 3.3. Statistical Analysis

All the data are presented as means ± standard error of the mean of at least three independent experiments. All the statistical analyses were performed using one-way analysis of variance (ANOVA) with Tukey’s post-hoc test followed by the Bonferroni post-test correction using GraphPad Prism 5.02 for Windows (GraphPad Software, San Diego, CA, USA). p Values < 0.05 were considered statistically significant.

## 4. Conclusions

In the IL-1β-stimulated human chondrocytes, the 50% ethanolic extract of Manchurian spikenard significantly inhibited IL-1β-induced expression of various inflammatory mediators such as IL-6, IL-8, MMP-1, MMP-13 and iNOS(NO), and the biomarkers of nociception such as COX-2 and its enzymatic product PGE2. However, IL-1β-induced expression of MMP-3 was not affected by the extract. Interestingly, continentalic acid in the extract, not kaurenoic acid, primarily mediated anti-inflammatory and anti-arthritic activity of the extract. The inhibitory activity of continentalic acid was compatible with that of the extract including an equivalent amount of continentalic acid. However, kaurenoic acid showed those inhibitory activities only at a concentration 10 times higher than that of continentalic acid. Anti-arthritic activities of Manchurian spikenard extract and continentalic acid were also verified in MIA-induced osteoarthritic rats. Taken together, continentalic acid, rather than kaurenoic acid, was most probably responsible for the anti-arthritic activity of Manchurian spikenard.

## Figures and Tables

**Figure 1 ijms-20-05488-f001:**
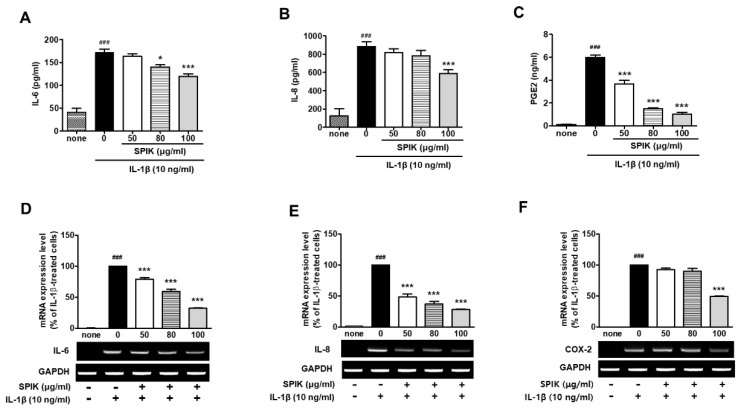
Effect of the 50% ethanolic extract of Manchurian spikenard on the IL-1β-stimulated protein and/or mRNA expression levels of IL-6 (**A**,**D**), IL-8 (**B**,**E**) and COX-2 (**F**), and PGE2 synthesis (**C**) in human osteoarthritis (OA) chondrocytes. SPIK, Manchurian spikenard extract; PGE2, prostaglandin E2; COX, cyclooxygenase; GAPDH, glyceraldehyde 3-phosphate dehydrogenase. ^###^
*p* < 0.001 vs. non-treated naïve cells (none); * *p* < 0.01 and *** *p* < 0.001 vs. the IL-1β-treated cells without treatments.

**Figure 2 ijms-20-05488-f002:**
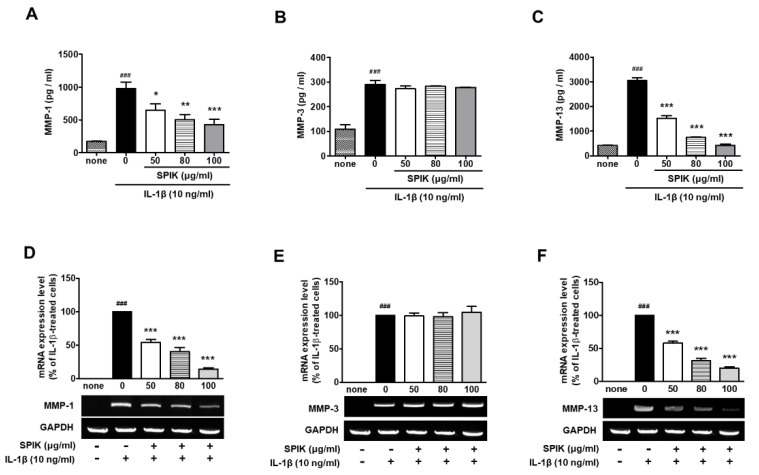
Effect of the 50% ethanolic extract of Manchurian spikenard on the IL-1β-stimulated protein and mRNA expression levels of MMP-1 (**A**,**D**), MMP-3 (**B**,**E**) and MMP-13 (**C**,**F**) in human OA chondrocytes. SPIK, Manchurian spikenard extract; MMP, matrix metalloproteinase; GAPDH, glyceraldehyde 3-phosphate dehydrogenase. ^###^
*p* < 0.001 vs. non-treated naïve cells (none); * *p* < 0.01, ** *p* < 0.005 and *** *p* < 0.001 vs. the IL-1β-treated cells without treatments.

**Figure 3 ijms-20-05488-f003:**
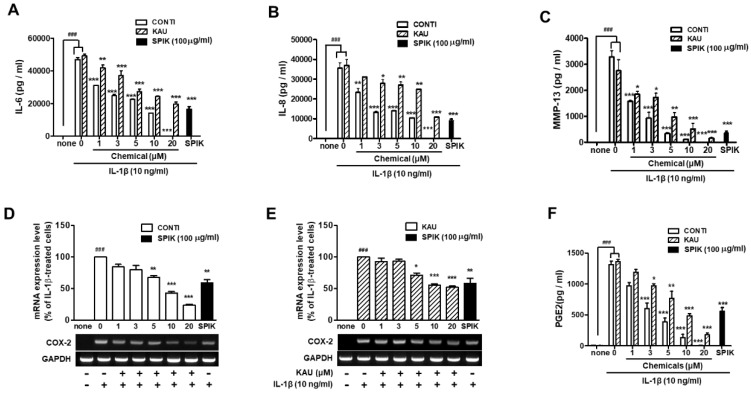
Effects of continentalic and kaurenoic acids on the IL-1β-stimulated protein or mRNA expression levels of IL-6 (**A**), IL-8 (**B**), MMP-13 (**C**) and COX-2 (**D**, continentalic acid; **E**, kaurenoic acid), and PGE2 production (**F**) in human OA chondrocytes. The 50% ethanolic extract of Manchurian spikenard (100 μg/mL) was used as a control (black bar in each graph). SPIK, Manchurian spikenard extract; CONTI, continentalic acid; KAU, kaurenoic acid; MMP, matrix metalloproteinase; COX, cyclooxygenase; PGE2, prostaglandin E2; GAPDH, glyceraldehyde 3-phosphate dehydrogenase. ^###^
*p* < 0.001 vs. non-treated naïve cells (none); * *p* < 0.01, ** *p* < 0.005 and *** *p* < 0.001 vs. the IL-1β-treated group without treatments.

**Figure 4 ijms-20-05488-f004:**
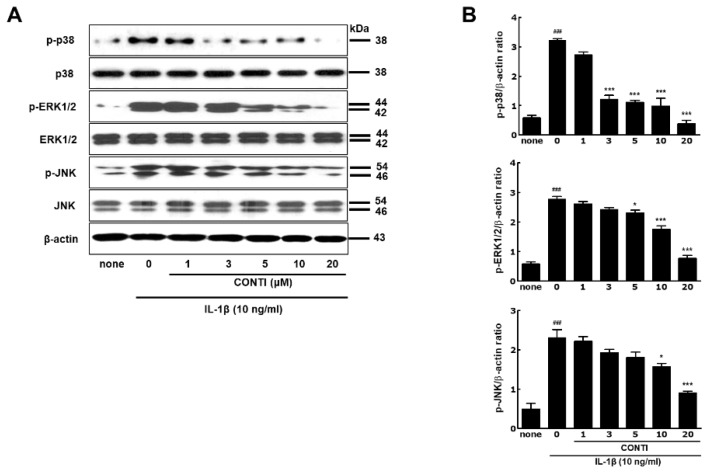
Western blot images (**A**) of *p*-p38, *p*-ERK and *p*-JNK MAP kinases, their bar graphs (**B**), and immunofluorescence image (**C**) of the nuclear translocation of NF-κB (p65) in IL-1β-stimulated human OA chondrocytes with the continentalic acid treatment. In Figure C, a’, b’ and c’ indicate non-treated naïve cells (none), IL-1β (10 ng/mL)-treated cells for 30 min (IL-1β), and IL-1β + CONTI (10 μM)-treated cells for 1 h (IL-1β + CONTI), respectively. ERK, extracellular signal-regulated kinase; p-ERK, phosphorylated ERK; JNK, jun N-terminal kinase. CONTI, continentalic acid. NF, nuclear factor. Original magnification: ×400 and scale bar: 40 μm. ^###^
*p* < 0.001 vs. non-treated group (none); * *p* < 0.01 and *** *p* < 0.001 vs. IL-1β-treated group without treatments.

**Figure 5 ijms-20-05488-f005:**
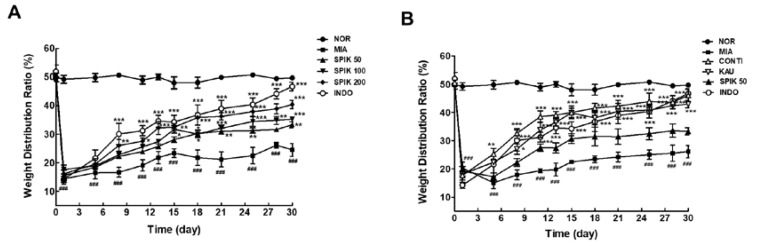
Behavioral assessments, indicated by weight distribution ratio of the per cent of weight carried on each hind leg, of anti−arthritic activities of the 50% ethanolic extract of Manchurian spikenard (**A**), and its chemical compounds such as continantalic and kaurenoic acids (**B**) in the monoiodoacetate (MIA)−induced monoarthritic rats. The spikenard extract was i.p. injected in Figure B being contrast with oral administration in Figure A. SPIK, Manchurian spikenard extract; CONTI, continentalic acid; KAU, kaurenoic acid; INDO, indomethacin. ^###^
*p* < 0.001 vs. non-treated normal group (NOR); * *p* < 0.05, ** *p* < 0.01, *** *p* < 0.001 vs. vehicle-treated MIA group without SPIK, CONTI or KAU treatment.

**Figure 6 ijms-20-05488-f006:**
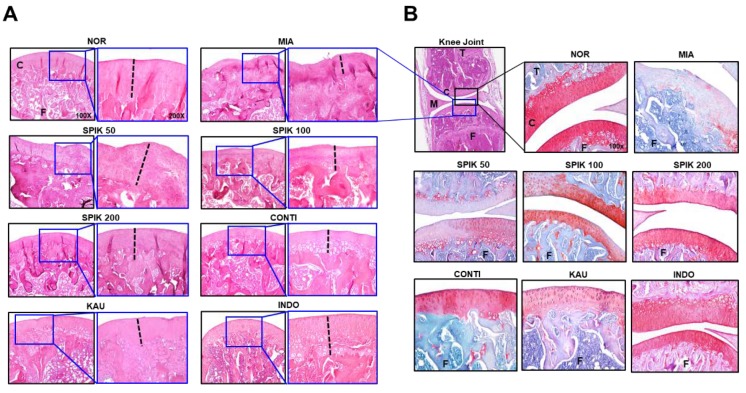
Histological images of rat’s knee joints using hematoxylin-eosin (H&E, **A**) and safranin O-fast green (S–F, **B**) staining solution. In Figure **A**, small blue squares in left photos (100×) are magnified to the right photos (200×) in each group. Dotted black lines in the right photos indicate the thickness of femur cartilage. Histological images in SPIK 50, SPIK 100 and SPIK 200 groups were from the rats orally administrated not intraperitoneally injected. In Figure B, first image shows the whole knee joint including femur (F), tibia (T), cartilage (C) and meniscus (M), and the blue and black squares indicate the anatomical regions for H&E and S–F staining, respectively. MIA, monoiodoacetate; SPIK, Manchurian spikenard extract; CONTI, continentalic acid; KAU, kaurenoic acid; INDO, indomethacin.

**Table 1 ijms-20-05488-t001:** Nucleotide sequences of primers for each gene and operating conditions of reverse transcription polymerase chain reaction (RT-PCR) analysis.

Gene (bp)		Nucleotide Sequence		Annealing Temp. (°C)/Cycles
human				
*GAPDH* (579)	sense	5′-ATC CCA TCA CCA TCT TCC AG-3′		58/30
	antisense	5′-CCT GCT TCA CCA CCT TCT TG-3′		
*IL-6* (150)	sense	5′-AGT TGC CTT CTT GGG ACT GA -3′		55/31
	antisense	5′-TCC ACG ATT TCC CAG AGA AC -3′		
*IL-8* (166)	sense	5′-GTT TTG CCA AGG AGT GCT AA -3′		55/31
	antisense	5′-CCA GAC AGA GCT CTC TTC CA -3′		
*COX-2* (158)	sense	5′-TGA GCA TCT ACG GTT TGC TG -3′		55/31
	antisense	5′-TGC TTG TCT GGA ACA ACT GC -3′		
*MMP-1* (388)	sense	5′-CCT AGC TAC ACC TTC AGT GG-3′		57/29
	antisense	5′-GCC CAG TAC TTA TTC CCT TT-3′		
*MMP-3* (365)	sense	5′-TCC CCC TGA CTC CCC TGA-3′		57/29
	antisense	5′-TCC TCA CGG TTG GAG GGA AA-3′ ′		
*MMP-13* (150)	sense	5′-TGA CCC TTC CTT ATC CCT TG-3′		57/29
	antisense	5′-ATA CGG TTG GGA AGT TCT GG-3′		
*iNOS* (320)	sense	5′-GCA TGT ACC CTC GGT TCT GT-3′		58/30
	antisense	5′-CAT GGT GAA CAC GTT CTT GG-3′		
mouse				
*GAPDH* (223)	sense	5′-AAC TTT GGC ATT GTG GAA GG-3′		58/30
	antisense	5′-ACA CAT TGG GGG TAG GAA CA-3′		
*IL-6* (159)	sense	5′-AGT TGC CTT CTT GGG ACT GA-3′		52/27
	antisense	5′-TCC ACG ATT TCC CAG AGA AC-3′		
*COX-2* (194)	sense	5′-AGA AGG AAA TGG CTG CAGAA 3′		55/28
	antisense	5′-GCT CGG CTT CCA GTA TTG AG -3′		
*iNOS* (199)	sense	5′-CCT CCT CCA CCC TAC CAA GT-3′		57/28
	antisense	5′-CAC CCA AAG TGC TTC AGT CA-3′		

T: thymine, A: adenine, C: cytosine, G: guanine, GAPDH: glyceraldehyde-3-phosphate dehydrogenase, COX: cyclooxygenase, MMP: matrix metalloproteinase, IL: interleukin, TNF: tumor necrosis factor, iNOS: inducible nitric oxide synthase.

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
