# Peer review of "Continentalic Acid Rather Than Kaurenoic Acid Is Responsible for the Anti-Arthritic Activity of Manchurian Spikenard In Vitro and In Vivo"

_ijms, 2019, doi:10.3390/ijms20215488_

Round 1

Reviewer 1 Report

General comment

The manuscript by Hong and coworkers reports an investigation of the anti-arthritic activity of the extract from the roots of Manchurian spikenard  as well as of its main components, on in-vitro and in-vivo models. Although in my opinion both the experimental approach and the interpretation of the experimental results are correct, the manuscript needs to be deeply revised and rewritten in numerous parts. Moreover, since the English is extremely poor, reading the manuscript is very difficult, especially for readers who are not experts in the field. The lack of details in all the sections (especially in the ‘materials and methods), in the figure legends, as well as the complexity of the figures, suggest the reorganization of the manuscript as a required process before the manuscript can be properly reviewed.

Introduction

In my opinion the introduction, in addition to information about Manchurian spikenard and its medical properties,  should report briefly, but clearly, both the aims of the work and a concise description of the models employed for the experiments. This would make clearer the comprehension of the rational of the investigation.

Figures

The are composed of to many graphs of very small size that result extremely confusing for the reader. Probably the presentation of most of the results as tables (reporting the data and eventually their  statistical significance) would facilitate their evaluation  by the reader.  Most of the images of the figures could be part of supplementary materials. With such a complexity in the figures, the legends do not make clear what they exactly report.

Results and Discussion

Due to the lack of information of  about the ‘aims ’ of the investigation, that are not clearly stated in the introduction,  and taking into account both the excessive brevity of the materials and methods and the poor clarity of the figures, the results section is extremely confusing. In my opinion, the results should be more concise and could be eventually combined with the discussion. Moreover, issues like the high dosages needed to highlight ant-arthritic effects of  the substances tested, both in vitro and especially in vivo, should be pointed out more clearly and discussed in deeper detail.  

Materials and methods

The materials and methods need to be reorganized and rewritten. The description of the experiments is extremely concise thus making extremely difficult their reproducibility by other scientists.  Since the number of different techniques employed is high, I would advice to prepare a supplementary information file reporting in detail the procedures adopted for the different experiments, as well as the correspondent bibliographic references.

Author Response

Reviewer 1

General comment

The manuscript by Hong and coworkers reports an investigation of the anti-arthritic activity of the extract from the roots of Manchurian spikenard as well as of its main components, on in-vitro and in-vivo models. Although in my opinion both the experimental approach and the interpretation of the experimental results are correct, the manuscript needs to be deeply revised and rewritten in numerous parts. Moreover, since the English is extremely poor, reading the manuscript is very difficult, especially for readers who are not experts in the field. The lack of details in all the sections (especially in the ‘materials and methods), in the figure legends, as well as the complexity of the figures, suggest the reorganization of the manuscript as a required process before the manuscript can be properly reviewed.

Response) According to the reviewer’s comment, Experimental Procedure (Materials & Methods) was totally corrected and highlighted in blue in the revised manuscript. Figure legends were also corrected (Several parts were deleted in each legend, and corrected. Figure 6 was split to two figures, etc.). Results and Discussion were properly combined (highlighted in red), and corrected (highlighted in blue) in the revised manuscript. The English in the revised manuscript was checked by at least two professional editors, both native speakers of English. For a certificate, please see: http://www.textcheck.com/certificate/PC3HSC

Introduction

In my opinion the introduction, in addition to information about Manchurian spikenard and its medical properties, should report briefly, but clearly, both the aims of the work and a concise description of the models employed for the experiments. This would make clearer the comprehension of the rational of the investigation.

Response) According to the reviewer’s comment, several texts in the introduction were corrected, and highlighted in blue in the revised manuscript. We corrected the description about Manchurian spikenard (highlighted in blue in the revised manuscript) also added the brief description of the rat model used in the present study.

Figures

They are composed of to many graphs of very small size that result extremely confusing for the reader. Probably the presentation of most of the results as tables (reporting the data and eventually their statistical significance) would facilitate their evaluation by the reader. Most of the images of the figures could be part of supplementary materials. With such a complexity in the figures, the legends do not make clear what they exactly report.

Response) According to the reviewer’s comment, ‘Supplementary Materials’ were created and several figures and table (Fig. 2, Fig. 5(A,B), Table 1) in previous manuscript were moved to the Supplementary file in the revised manuscript. And Figure 6 were split to two figures (Fig. 6 and 7), and the schematic diagrams of in vitro and in vivo experimental schedules were created as a supplementary figure S2 for clarity in the revised manuscript. We also corrected all figure legends, and highlighted in blue in the revised manuscript.

Results and Discussion

Due to the lack of information of  about the ‘aims ’ of the investigation, that are not clearly stated in the introduction,  and taking into account both the excessive brevity of the materials and methods and the poor clarity of the figures, the results section is extremely confusing. In my opinion, the results should be more concise and could be eventually combined with the discussion. Moreover, issues like the high dosages needed to highlight ant-arthritic effects of the substances tested, both in vitro and especially in vivo, should be pointed out more clearly and discussed in deeper detail.  

Response) According to the reviewer’s comment, Results and Discussion were properly combined (highlighted in red), and corrected (highlighted in blue) in the revised manuscript. And Experimental Procedure (Materials & Methods) were properly reorganized, supplemented, corrected and highlighted in blue color in the revised manuscript. We also added a paragraph describing the dose effects of both substances in terms of anti-inflammatory and anti-arthritic activity (highlighted in blue in page 10 in the revised manuscript), according to the reviewer’s comment.

Materials and methods

The materials and methods need to be reorganized and rewritten. The description of the experiments is extremely concise thus making extremely difficult their reproducibility by other scientists.  Since the number of different techniques employed is high, I would advice to prepare a supplementary information file reporting in detail the procedures adopted for the different experiments, as well as the correspondent bibliographic references.

Response) According to the reviewer’s comment, Experimental Procedure (Materials & Methods) was totally corrected and supplemented, and highlighted in blue in the revised manuscript. And ‘Supplementary Materials’ were created and several figures and table (Fig. 2, Fig. 5(A,B), Table 1) in previous manuscript were moved to the Supplementary file in the revised manuscript.

Reviewer 2 Report

The manuscript presents very interesting results regarding the therapeutical effect of continentalic and kaurenoic acids for osteoarthritis treatment. Both acid are presented in the extract of Manchurian spikenard roots and they were used for the treatment of OA. According to the carreful search on Scopus, this kind of investigation was conducted only once and only this manuscript describing in a very scientific manner the role of two acids for the OA treatment. Although this work has very high importance of OA treatment community, its quality can be even more improved

1) The quality of Figures. The Figures are overloaded and too abundant. Authors should better describe their message on the Figures

2) The histology results are not enough clear. Also better discussion of the histology results is necessary

3) The Figure captions are too long and they are separated from the Figures (on different pages), please format the text and Figures.

Author Response

The manuscript presents very interesting results regarding the therapeutical effect of continentalic and kaurenoic acids for osteoarthritis treatment. Both acid are presented in the extract of Manchurian spikenard roots and they were used for the treatment of OA. According to the carreful search on Scopus, this kind of investigation was conducted only once and only this manuscript describing in a very scientific manner the role of two acids for the OA treatment. Although this work has very high importance of OA treatment community, its quality can be even more improved

1. The quality of Figures. The Figures are overloaded and too abundant. Authors should better describe their message on the Figures

Response) According to the reviewer’s comment, quality of figures were improved and all figures were totally reorganized in the revised manuscript. ‘Supplementary Materials’ were created and several figures and table (Fig. 2, Fig. 5(A,B), Table 1) in previous manuscript were moved to the Supplementary file in the revised manuscript. And Figure 6 were split to two figures (Fig. 6 and 7), All figure legends were also corrected (Several parts were deleted in each legend, and corrected. Figure 6 was split to two figures, etc.).

2. The histology results are not enough clear. Also better discussion of the histology results is necessary

Response) According to the reviewer’s comment, the description about histological results were added and corrected, and highlighted in blue in page 13-14 (line 303-311) in the revised manuscript.

3. The Figure captions are too long and they are separated from the Figures (on different pages), please format the text and Figures.

Response) According to the reviewer’s comment, all figure legends were properly corrected (Several parts were deleted in each legend, and corrected. Figure 6 was split to two figures, etc.). We also reorganized and corrected whole text and figures in the revised manuscript.

Round 2

Reviewer 1 Report

I appreciate that the authors ameliorated the English language and that they improved the ‘Materials and Methods’  and set up a Supplementary material section. Nevertheless I think that still some major points still need to be considered, mostly concerning the organization of the manuscript that also in the present version results confusing .

Introduction.

I think that although the new texts (in blue) are pertinent, they cannot just be added, but rather integrated into the text that, in my opinion, should be rewritten according to this suggestion.

Figures.

They are still to many and to complex. Especially  those reporting different graphs or photo of the same experiment: a selection should be done and the remaining should be moved to the supplementary material section.

Results and discussion

The simplification of the figures will certainly result in a shorter ‘result and Discussion section’, that now is redundant and confusing.

Material and methods

Now this section is complete, but again I would suggest to move almost all the detailed descriptions of the experiments to the Supplementary section.

In conclusion I think that although the experimental work and the conclusions are correct, the manuscript must be more concise in order to make more evident aims and scope of the investigation as well as the quality of the experimental approach adopted. I suggest a shorter and simplified manuscript and a well organized Supplementary section containing the detailed Materials and a large prte of the figures accompanied by comments. A major revision of the manuscript, in my opinion is still needed.

Author Response

I appreciate that the authors ameliorated the English language and that they improved the ‘Materials and Methods’ and set up a Supplementary material section. Nevertheless I think that still some major points still need to be considered, mostly concerning the organization of the manuscript that also in the present version results confusing.

Introduction.

I think that although the new texts (in blue) are pertinent, they cannot just be added, but rather integrated into the text that, in my opinion, should be rewritten according to this suggestion.

Response) According to the reviewer’s comment, ‘Introduction’ was downsized a little, and corrected and highlighted in blue in the revised manuscript.

Figures.

They are still to many and to complex. Especially those reporting different graphs or photo of the same experiment: a selection should be done and the remaining should be moved to the supplementary material section.

Response) According to the reviewer’s comment, Fig 1, Fig 2(A and G), Fig 3(G), Fig 4(A) were moved to ‘Supplementary Materials’, and Figure S2 of chondrocyte cell viability (toxicity), Figure S3 of NO and its synthesizing enzyme iNOS, and Figure S6 of HPLC chromatogram were created in the revised manuscript. Figures were reorganized and downsized.

Results and discussion

The simplification of the figures will certainly result in a shorter ‘result and Discussion section’, that now is redundant and confusing.

Response) According to the reviewer’s comment, ‘Results and Discussion’ was downsized (many paragraph were deleted and re-wrote), and highlighted in blue in the revised manuscript.

Material and methods

Now this section is complete, but again I would suggest to move almost all the detailed descriptions of the experiments to the Supplementary section.

Response) According to the reviewer’s comment, the descriptions of HPLC analysis, cell viability assay and NO assay were moved to ‘Supplementary Materials in the revised manuscript.

In conclusion I think that although the experimental work and the conclusions are correct, the manuscript must be more concise in order to make more evident aims and scope of the investigation as well as the quality of the experimental approach adopted. I suggest a shorter and simplified manuscript and a well organized Supplementary section containing the detailed Materials and a large part of the figures accompanied by comments. A major revision of the manuscript, in my opinion is still needed.

Response) According to the reviewer’s comment, the manuscript was downsized and reorganized, especially several figures were moved to ‘Supplementary Materials’, several paragraphs were removed and rewrote, and the structure of Figures was totally changed in the revised manuscript. ‘Conclusions’ were also downsized and rewrote (highlighted in blue in the revised manuscript).

Round 3

Reviewer 1 Report

In this revised version, in my opinion the manuscript can be considered for publication.